# Network-Based In Silico Analysis of New Combinations of Modern Drug Targets with Methotrexate for Response-Based Treatment of Rheumatoid Arthritis

**DOI:** 10.3390/jpm13111550

**Published:** 2023-10-29

**Authors:** Marjan Assefi, Kai-Uwe Lewandrowski, Morgan Lorio, Rossano Kepler Alvim Fiorelli, Stefan Landgraeber, Alireza Sharafshah

**Affiliations:** 1Marie Curie Science Research Center, Greensboro, NC 27407, USA; massefi@aggies.ncat.edu; 2Center for Advanced Spine Care of Southern Arizona, 4787 E Camp Lowell Drive, Tucson, AZ 85712, USA; business@tucsonspine.com; 3Department of Orthopaedics, Fundación Universitaria Sanitas, Bogotá 111321, Colombia; 4Department of Orthopedics, Hospital Universitário Gaffre e Guinle, Universidade Federal do Estado do Rio de Janeiro, Rio de Janeiro 21941-590, RJ, Brazil; 5Advanced Orthopaedics, 499 E. Central Pkwy, Ste. 130, Altamonte Springs, FL 32701, USA; mloriomd@gmail.com; 6Department of General and Specialized Surgery, Gaffrée e Guinle University Hospital, Federal University of the State of Rio de Janeiro (UNIRIO), Rio de Janeiro 22290-240, RJ, Brazil; fiorellirossano@hotmail.com; 7Klinik für Orthopädie und Orthopädische Chirurgie Gebäude 37, EG, Zimmer 56, 66421 Homburg, Germany; stefan.landgraeber@uks.eu; 8Cellular and Molecular Research Center, School of Medicine, Guilan University of Medical Sciences, Rasht P.O. Box 4144654839, Iran

**Keywords:** Rheumatoid arthritis, Inflammatory arthritis, DMARDS, TLR4 receptor inhibition, JAK inhibitors, new targeted therapy

## Abstract

Background: Methotrexate (MTX), sulfonamides, hydroxychloroquine, and leflunomide have consistently resulted in remission with relatively mild to moderate adverse effects in patients with rheumatoid arthritis (RA). Modern medications outperform traditional treatments in that they target the pathological processes that underlie the development of RA. Methods: Following PRISMA guidelines, the authors accomplished a systematic review of the clinical efficacy of RA drugs, including the biologics such as Tumor Necrosis Factor-alpha inhibitors (TNF-α i) like Etanercept, Infliximab, Golimumab, and Adalimumab, kinase inhibitors (JAK inhibitors including Baricitinib and Tofacitanib), SyK inhibitors like Fos-tamatinib, MAPK inhibitors such as Talmapimod, T-cell inhibitors (Abatacept), IL6 blockers (Tocilizumab), and B cells depleters (Rituximab). These drugs have been found to increase remission rates when combined with MTX. A bioinformatics-based network was designed applying STRING-MODEL and the DrugBank database for the aforementioned drugs and MTX and, finally, employed for this systematic review. Results: Current research demonstrates that non-TNF-α inhibitor biologicals are particularly helpful in treating patients who did not respond well to conventional medications and TNF-α inhibitors. Despite being effective, these innovative drugs have a higher chance of producing hazardous side effects. The in silico investigations suggested an uncovered molecular interaction in combining MTX with other biological drugs. The STRING-MODEL showed that DHFR, TYMS, and ATIC, as the receptors of MTX, interact with each other but are not connected to the major interacted receptors. Conclusions: New game-changing drugs including Mavrilimumab, Iguratimod, Upadacitinib, Fenebrutinib, and nanoparticles may be crucial in controlling symptoms in poorly managed RA patients. Emerging therapeutic targets like Toll-like 4 receptors, NLRP3 inflammasome complexes, and mesenchymal stem cells can further transform RA therapy.

## 1. Introduction

Rheumatoid arthritis (RA) is an autoimmune inflammatory systemic disease that predominantly affects joints and eventually results in joint damage and disability [1,2]. Approximately 1.3 million individuals in the United States suffer from RA, accounting for 0.6% to 1% of the adult population [3,4]. Patients with RA may experience negative long-term consequences such as disability, a decline in quality of life, and increased mortality [5]. Work disability is a significant outcome of RA, and it has been claimed that the indirect cost of the disease owing to lost workability is almost three times more than the cost of treating the disease [6]. As a result, RA treatment aims to avoid injury to the structure and function of the joints, making early, effective treatment necessary to manage inflammation and maintain quality of life [7]. RA begins with the involvement of smaller joints and progresses to bigger joints and the skin, eyes, heart, kidneys, and lungs [8]. Bone erosion joint space narrowing, subluxation and luxation, and abnormalities are frequently caused by the destruction of the bone and cartilage in joints [9]. Morning stiffness in the afflicted joints, weariness, fever, weight loss, sensitive, swollen and warm joints, and rheumatoid nodules under the skin are common symptoms [10,11]. In RA, anti-inflammatory cytokines counteract the persistent activation of innate and adaptive immune cells [12]. The presence of activated NF-B in the synovial tissue of RA patients has long been known [13]. By functioning in a variety of cell types, NF-B contributes to the pathophysiology of RA. It first promotes the activation of pro-inflammatory cytokines in monocytes and macrophages, including TNF, IL-1, and IL-6. The majority of these cytokines have the ability to activate NF-B in fibroblasts and innate immune cells, which further expands inflammation [14]. Secondly, NF-B supports self-reactive B cell survival and Th17 differentiation, both of which are cell types that are strongly involved in the development of RA [15,16]. For unexplained reasons, RA affects more women than men and often begins during middle age, between the ages of 40 and 50. Age, lifestyle, and weight are other risk factors. Notably, RA often manifests as remission and exacerbation [17].

The treatment method for RA includes pharmaceuticals, weight-bearing exercises, and patient education [18]. The overarching objective of first-line treatment is to alleviate pain and reduce inflammation. Nonsteroidal anti-inflammatory drugs (NSAIDs) such as naproxen, ibuprofen, and etodolac are instances of fast-acting medications [17,19]. A second-line RA therapy aims to achieve remission by slowing or preventing the progression of joint degradation and deformity; these medicines are called slow-acting since they take weeks to months to become effective [20]. These disease-modifying anti-rheumatic drugs (DMARDs) can also lower the incidence of lymphoma, which is associated with RA [20]. Methotrexate (MTX) is used as both a first-line and the initial second-line drug in RA treatment (also considered an anchor drug) and is an analog to folic acid that impairs the metabolism of purines and pyrimidines and, subsequently the synthesis of amino acids and polyamine [21]. The major side effects of MTX are reported as carcinogenicity, gastrointestinal, hepatotoxicity, pulmonary toxicity, nephrotoxicity, hematologic toxicity, and infection [22]. Because of the lower dose, life-threatening adverse effects are rare in the RA therapy of MTX, although some problems may occur and produce serious consequences independent of the amount of dose, including hepatotoxicity [23,24].

Leflunomide, a conventional synthetic DMARD (csDMARD) and an oral medication, inhibits the formation of ribonucleotide uridine monophosphate pyrimidine [25]. It decreases RA signs and delays the progression of the disease [26]. Although it is best used in combination with MTX, it can be used alone if patients do not react to MTX. Hypertension, GI discomfort, liver damage, leukopenia, interstitial lung disease, neuropathy, dermatitis, and bone marrow destruction are all identified side effects [27]. Another drug in this group is hydroxychloroquine. This antimalarial drug can be used for long-term RA treatment, as it decreases the secretion of mono-cyte-derived proinflammatory cytokines. Common side effects include problems in the Gastrointestinal (GI) tract, skin, and central nervous system. The eyes, in particular, can be affected when this drug is taken at high doses. Patients treated with this medication require routine consultation with an ophthalmologist [28]. The last drug mentioned is sulfasalazine (Azulfidine), a DMARD typically used to treat RA when combined with anti-inflammatory medications [18]. According to Smedegard and Bjork, the activity of sulfasalazine in RA is most likely mediated through some of the immunomodulatory and anti-inflammatory functions [29].

Biologic medications, more modern, targeted treatments that are often added to the background of traditional disease-modifying antirheumatic drugs, have recently revolutionized the treatment of RA [30,31]. Furthermore, the introduction of these medications has broadened the therapeutical approach to RA due to the availability of biosimilar Tumor Necrosis Factor inhibitors (TNFi), target-cells, interleukin-6, or T-cell co-stimulation [32,33].

TNF is a messenger protein that stimulates joint inflammation [34]. TNFi are biological drugs including etanercept, infliximab, adalimumab, golimumab, and certolizumab pegol, which work quickly to relieve symptoms by preventing the recruitment of the cells that produce inflammation [35,36,37]. If other second-line drugs are ineffective, these are advised. They are often used in combination with other DMARDs, especially MTX. TNFi is contraindicated in patients with congestive heart failure or demyelinating diseases [38]. Furthermore, Anakinra is a drug that works by binding to IL-1, a chemical messenger of inflammation. It can be combined with other DMARDs or as a monotherapy [39,40,41]. Rituximab is helpful in RA because it depletes the B-cells responsible for inflammation and the production of abnormal antibodies [42]. Typically used in treating lymphoma, rituximab can be used in cases of RA where TNFi has failed [43]. A biological drug called abatacept functions by inhibiting T-cell activation [44]. Patients who have not responded well to usual DMARDs are treated with it. A biological drug called tocilizumab reduces inflammation by preventing IL-6, a chemical messenger [45]. Baricitinib and upadacitinib have both been approved and are used in the clinical treatment of RA. The first drug in the JAKi class to receive approval for the treatment of RA in the European Union (EU) is baricitinib, an oral selective and reversible JAK1/JAK2 inhibitor. This is due to evidence of its superior efficacy when compared with placebo and tumor necrosis factor inhibitors (TNFis) in people with an insufficient response (IR) to csDMARDs, like MTX [46,47,48]. Upadacitinib, another JAK inhibitor with better sensitivity for JAK1, has documented a good benefit: a risk profile for individuals who have not responded well to bDMARDs and csDMARDs [49]. Last but not least, tofacitinib (Xeljanz) acts by suppressing inflammatory enzymes known as Janus kinases 1 and 3 (JAK1 and JAK3) within cells. As a result, it is known as a JAK inhibitor [50]. This drug is prescribed for people who have not responded well to MTX. Tofacitinib can be given alone or with MTX; however, it should not be combined with conventional biologic medicines or other potent immunosuppressants [51]. In this systematic review article, the authors aimed to evaluate the efficacy and side-effect profile of the various traditional treatment modalities and emerging therapies and their impact on the adult RA population for improved patient-centered outcomes [52]. Thus, in this review, the authors employed new network-based bioinformatics analyses to uncover newer, more targeted treatment strategies with modern DMARDs and MTX combination therapies.

## 2. Materials and Methods

### Study Strategy

Following prisma guidelines [1], this review systematically searched Rheumatoid Arthritis in PubMed (https://pubmed.ncbi.nlm.nih.gov/ accessed on 1 September 2023) and found 69,853 published papers and books from 1876 to 2023. By searching Rheumatoid Arthritis drugs, the results decreased to 18,253 entries from 1947 to 2023. In the next stage, by adding biologicals, the results declined to 9038 numbers. These search strategies revealed 2344, 376, and 43 publications for TNF-α, JAK, and MAPK, respectively. “Rheumatoid Arthritis drugs biologicals combined with MTX” represented 866 related references filtered to 236 publications with significant reports (Figure 1) [40]. “Novel drugs” were searched, and results indicated 10 publications for Mavrili-mumab, 11 papers for Igarutide, 3 for Fenebrutinib, and 21 for nanoparticles. The results for emerging therapeutic targets like Toll-like 4 receptors were 2 publications. For NLRP3 inflammasome complexes, we found 13 publications, and for the use of mesenchymal stem cells, 50 PubMed-indexed papers. Additionally, another search strategy focused on the first-line treatment of RA. The PubMed database was filtered in the last five years from 2018 to 2023 with “First-line” and “Rheumatoid Arthritis” as keywords. Three categories were identified, including RA and MTX, RA and NSAIDs, and RA and other medications. Notably, 171 publications were found, 73 documenting MTX, and 97 papers reported other drugs and various biologics when excluding MTX. Only one paper suggested NSAIDs as the first-line RA therapy.

The authors searched the DrugBank database to extract the targets of RA drugs by employing a bioinformatics-based network utilizing STRING-MODEL for the drugs mentioned above and MTX to facilitate their systematic review and to identify novel more targeted RA treatments by combining MTX with modern RA drugs.

## 3. Results

### 3.1. Results from the Literature

According to the American College of Rheumatology (ACR) guidelines in 2021, MTX is the first-line treatment for RA. TNFi was the first class of bDMARD which has been approved for the treatment of RA and it is reported that it can initiate a positive impact on patients’ quality of life [53]. Based on the current clinical guidelines, MTX monotherapy is significantly preferred to bDMARD or tsDMARD monotherapy, MTX plus a non-TNFi bDMARD, or targeted synthetic DMARDs (tsDMARDs) in DMARD-naive patients with moderate-to-high disease activity. Notably, the term “difficult-to-treat RA” (D2T RA) has been used to describe a subset of patients whose disease activity remains uncontrollable regardless of the use of two or more biological DMARDs or tsDMARDs with different mechanisms of action [54,55].

If necessary, the treatment-to-target strategy may avoid RA-related impairments by lowering disease activity by at least 50% within three months and attaining remission or low disease activity by six months. When MTX and glucocorticoids are used together, 40% to 50% of patients can achieve remission or have minimal disease activity. When the first MTX and glucocorticoid regimen fails, the successive administration of targeted medicines, such as biologic drugs or Janus kinase inhibitors in combination with MTX, has allowed up to 75% of patients to attain the therapeutic target over time [56].

In cases of high disease activity, the presence of autoantibodies, early erosions, or the failure of two conventional DMARDs, the addition of bDMARD or a targeted synthetic JAK kinase inhibitor to a conventional synthetic (csDMARD) is necessary [20]. In patients with RA, beginning a second non-TNFi bDMARD after a first non-TNFi bDMARD failed provided a moderate response; however, the advantage was somewhat more significant when the second bDMARD was TNFi [57]. Therefore, it is better to switch to a bDMARD from a different class than to switch to a bDMARD from the same class; for example, if the first bDMARD is a TNFi, it is preferable for the second bDMARD to be a non-TNFi. In contrast to the more expensive bDMARDs, more economical biosimilar drugs are developing [56,58].

For individuals who have not responded to biological treatment, newer drugs such as mavrilimumab (an antibody against GM-CSF), rituximab (an anti-CD20 antibody), and tocilizumab (which inhibits IL6) can be utilized [59,60,61]. Pulse Wave Velocity (PWV), a risk factor for cardiovascular disease, was lowered in RA patients receiving tocilizumab and rituximab [60]. Janus Kinase (JAK) inhibitors (tofacitinib, baricitinib, upadacitinib, and filgotinib), spleen tyrosine kinase (SyK) inhibitors such as fostamatinib and mitogen-activated protein kinase (MAPK) inhibitors (talmapimod) show promising results in the treatment of RA patients who have failed to respond adequately to csDMARD and bDMARD [61,62,63,64]. In a study comparing the efficacy of tofacitinib monotherapy to MTX monotherapy in MTX-naive RA patients, the tofacitinib group had a significantly smaller mean difference in their modified total Sharp score progression, with 63.2% of patients receiving tofacitinib. In comparison, 12.0% of patients in the MTX group achieved ACR 70 response throughout the six-month study. However, the tofacitinib group was associated with more reports of herpes-zoster infection, cancer, high cholesterol, and elevated creatinine [33]. In addition, the research found that clinical responses to tofacitinib were better in bDMARD-naive patients than in bDMARD-IR individuals [65]. Tofacitinib had higher rates of major adverse cardiovascular events (MACE) and cancer (3.4% and 4.2%, respectively) than TNFi (2.5% and 2.9%, respectively). Tofacitinib considerably increases the risk of opportunistic infections such as herpes zoster and tuberculosis compared with TNFi [66].

Upadacitinib was more effective than abatacept at lowering Disease Activity Score 28 with C-Reactive protein (DAS28-CRP) levels and achieving clinical remission, but its side effects are also concerning [67]. A randomized phase III clinical trial on filgotinib versus placebo or adalimumab in MTX non-responders showed filgotinib (at a dose of 200 mg) was superior to placebo in improving symptoms, physical function, and radiographic disease progression and non-inferior to adalimumab in terms of DAS28-CRP < or equal to 3.2 at week 12 [68]. Iguratimod and nanoparticles, two novel game-changers, have demonstrated promising outcomes [69,70]. Iguratimod has significantly decreased DAS28-CRP levels over a six-month usage. As a result, iguratimod may be a new therapeutic option for RA [71]. Fenebrutinib, a Bruton tyrosine kinase inhibitor, had a promising favorable outcome in MTX non-responders comparable to Adalimumab in a randomized, double-blind phase II trial (ANDES Study) [72].

Despite significant progress in understanding the cellular and molecular pathways that cause inflammation in RA, the disease’s origin remains unknown. As mentioned, RA is a multifactorial disease associated with hereditary and environmental causes. Since HLA-DRB1 was identified as a susceptible gene in RA, more than 30 additional loci, such as PADI4, PTPN22, and FCRL3, have been demonstrated to contain genetic variants which can promote the disease [73]. Cigarette smoking is also proposed as a primary environmental trigger for RA, particularly in genetically susceptible people, which is corroborated by findings from animal models [74,75,76]. Furthermore, infection and tissue damage have been linked to the initiation of RA inflammation. Toll-like receptors (TLRs), important in detecting infection and damage, have been postulated to promote inflammation in RA within the last decade [77]. Several studies provided convincing evidence for the existence of TLRs in human synovial tissues. Even in the early phases of RA, TLR3 and TLR4 are abundantly expressed in human synovial fibroblasts [78]. TLR2 expression was shown to be high in areas of cartilage and bone destruction [79]. TLR2, TLR3, and TLR7 expressions were considerably higher in RA synovial fibroblasts (RASFs) than in healthy controls or individuals with non-inflammatory arthritis [69,80,81,82]. Multiple studies in RA patients report a dysfunctional TLR response and an abnormal presence of endogenous TLR ligands in synovial fluid and serum that may contribute to the onset of a chronic inflammatory state. As a result, promising treatments that target TLRs to treat and prevent RA are emerging [83]. In the long run, using nicotinic receptor agonists and 2-methoxy estradiol may also result in symptom amelioration and disease modification in patients who have rheumatoid arthritis [84,85]. There are several strategies for preventing TLR-induced inflammatory responses.

Soluble decoy receptors and neutralizing antibodies are two mechanisms for modulating the interaction between a receptor and a ligand. It is also possible to limit the production of endogenous ligands since DAMPs have been associated with TLR activation in RA. The in vivo usage of siRNA targeting TLR3 has lately been reported. The down-regulation of TLR expression might also be beneficial [86]. Another strategy is that TLR activation necessitates receptor dimerization, which occurs before TLR activation. Additionally, compounds acting as intermediary proteins downstream of TLR signaling might be effective candidates. Prior reports have evaluated and reported on the therapeutic targeting of TLRs for many pathological diseases, which include oncogenesis, infection, and autoimmune diseases [87,88,89].

In vitro, lipopolysaccharide (LPS)-stimulated fibroblast-like synoviocytes displayed decreased IL-6, IL-8, MMP-1, and VEGF expression by a TLR4 inhibitor known as TAK-242 (resatorvid). TAK-242 inhibited the mobilization of Nuclear Factor κB (NF-κB) and Activator Protein 1 (AP-1) into the nucleus, reduced the expression of essential transcription factors (NF-κB-p65 and AP-1) in the synovial tissue of adjuvant-induced arthritis (AIA) rats, and improved local joint inflammation and bone damage in AIA rats when studied in adjuvant-induced arthritic rats. Therefore, resatorvid has the potential to emerge as an innovative and potent treatment for RA [90]. The role of nucleotide-binding domain and leucine-rich repeat pyrin-containing protein-3 (NLRP3) inflammasome (members of the NOD-like receptor family) in the pathophysiology of RA has been the focus of numerous studies to date. The regulation of IL-1 beta and its release were linked to the NLRP3 inflammasome [91,92,93]. The joints of RA patients were found to have high expressions of components of the NLRP3 inflammasome [94].

Mesenchymal stem cells (MSCs) are a viable alternative treatment method because they inhibit the growth of effector memory T cells, discovered in high concentrations in RA patients’ serum and synovial fluid. MSCs can control the inflammatory response in RA by reducing effector T cell proliferation and generating inflammatory cytokines such as Interferon (IFN), IL-4, and IL-17 [95]. Extensive research on the therapeutic potential of MSCs for the treatment of RA has been conducted using the collagen-induced arthritis (CIA) model in mice. MSCs derived from the umbilical cord were discovered to be more suited for medical usage than stem cells derived from human exfoliated deciduous teeth (SHED) and bone marrow-derived MSCs (BMSCs) [96]. Extracellular vesicles (EVs) generated by MSCs have recently been identified as significant paracrine messengers in the healing process, impacting the surrounding microenvironment with anti-inflammatory actions. Cosenza and colleagues provided the first proof suggesting that MSC-derived EVs can have an immunomodulatory influence in a preclinical model of RA [94]. In this regard, independent of the priming state of the MSCs employed for isolation, the anti-inflammatory activity of MSC-derived EVs on T and B lymphocytes was demonstrated. While preliminary studies have shown that MSC treatment has therapeutic promise in RA, optimizing them for clinical application is a constant issue and genetically modified MSCs are being suggested to enhance their therapeutic potential [97,98,99,100]. Regardless of the tissue origin of the MSCs or the delivery system, MSC-based treatment can reduce the level of arthritic inflammation in most experimental models of RA by up to 30%. Additionally, early MSC infusion during the disease’s induction phase and an average cell dosage of 2–3 × 106 MSCs per animal utilizing either one or many MSC infusions have effectively regulated experimental RA [101]. Figure 2 summarizes all the RA treatments as mentioned above.

Rheumatoid arthritis is a multifactorial disease with different treatment modalities [102]. Each treatment has a variable effect on the disease process. Even though we have a variety of options, a patient cannot be prescribed any class of medication with specific outcomes. The data suggest that MTX is the first-line drug for RA. MTX and glucocorticoid combination treatment resulted in remission in 40 to 50% of patients (Figure 3).

### 3.2. Results from Bioinformatics Analyses

In RA patients with high disease activity, the presence of autoantibodies, and the erosion of joints, the addition of biological DMARDs and targeted synthetic JAK kinase inhibitors proved beneficial [20]. In patients for whom biological therapy has failed, the use of drugs like mavrilimumab (an antibody against GM-CSF), rituximab (an Anti CD20 antibody), and tocilizumab (rheumatoid arthritis is a multifactorial disease that blocks IL6) were encouraging [103,104].

A comparison between the efficacy of tofacitinib and MTX in MTX naïve RA showed a significantly smaller mean difference in their modified total Sharp score progression in the tofacitinib group with more adverse reactions, like herpes-zoster, cancer, high cholesterol, and high creatinine [33]. Tofacitinib utilization in the bDMARDs naïve group revealed better clinical responses than in the bDMARDs resistant group. Compared with TNFi, tofacitinib showed a higher incidence of MACE and cancer [66]. According to the DrugBank database (https://go.drugbank.com/), which showed a simplified view of the mechanism of action of MTX and other drugs, the current review designed a molecular network based on the MTX in combination with 11 FDA-approved drugs, including etanercept, infliximab, golimumab, adalimumab, baricitinib, tofacitinib, fostamatinib, talmapimod, abatacept, tocilizumab, and rituximab. The candidate targets of these drugs were obtained from DrugBank, considering their known pharmacological actions and trusted molecular evidence. The final targeted genes, which are protein-coding, were as follows: TYMS, ATIC, and DHFR for MTX; TNF, LTA, FCGR1A, FCGR2A, FCGR2B, FCGR2C, FCGR3A, and FCGR3B for etanercept; TNF for golimumab and adalimumab; JAK1, JAK2, JAK3, and TYK2 for baricitinib; SYK for fostamatinib; MAPK14 for talmapimod; CD80 and CD86 for abatacept; IL6R for tocilizumab; and MS4A1 for rituximab. Interestingly, the STRING-MODEL indicated no link between MTX targets (TYMS, ATIC, and DHFR genes) and other drugs. This means an unknown molecular mechanism should associate the MTX signaling pathway with the different pathways in which combined drugs are involved (Figure 4). This finding may shed light on future studies of RA and drug administration. Also, this suggestion may result in novel drug discovery, and the network of protein-coding genes involved in the treatment of RA can be a reference to design new NGS-based panels for personalized medicine via Whole Exome Sequencing (WES) and Whole Genome Sequencing (WGS).

## 4. Discussion

RA is a debilitating autoimmune disease that first affects joints before spreading to the skin, eyes, heart, kidneys, and lungs, causing multi-systemic complications. Consequently, treating RA as soon as possible is essential to avoid joint degeneration, retain the quality of life, and prevent disability. Weight-bearing exercises, patient education, NSAIDS, and corticosteroids are employed as first-line treatments for RA to decrease inflammation and offer symptomatic relief for acute episodes. With a considerable decrease in DAS28-CRP levels over six months with iguratimod use, new therapy strategies, including nanoparticles and iguratimod, have shown promise. Drug-loaded nanoparticle carriers have several benefits over conventional medications, such as enhanced insoluble drug delivery, targeted detection of the target cells, reduced systemic side effects, protection from drug degradation, controlled drug release, the promotion of drug transport across the biomembrane, and merged diagnostic tools as theranostic agents. Although several drug-loaded nanocarriers have been designed for testing in animal and cell-based investigations, there have been few clinical trials regarding their use in RA management. Clinical studies are costly, and unidentified adverse effects must first be found and investigated in Phase I and II trials. Because of their tiny size, which results in a proportionally increased surface area, nanomaterials are expected to cause more severe adverse impacts on organisms than materials with larger-sized particles. As a result, researchers are focusing on both the adverse effects that nanoparticles may have on human health and the environment and their therapeutic advantages. It is essential to exercise caution while attempting to balance the risk of tissue damage or toxicity with the effectiveness of anti-RA medication. The ingredients used to make anti-RA nanotherapeutics should be nontoxic, biodegradable, and biocompatible for human applications [69,70]. In MTX nonresponders, fenebrutinib showed promising outcomes comparable to adalimumab [72]. A dysfunctional Toll-like receptor response and endogenous TLR ligands in RA patients led to a focus on the disease’s pathophysiology, with resatorvid, a newer drug, targeting TLR-4 receptors [90]. Studies on the pathophysiology of inflammatory and autoimmune responses associated with RA were further prompted by the association of the NLRP3 inflammasome in the synovial joints of RA patients [90,91,92,93]. Through their immunomodulatory effect on effector T cells, mesenchymal stem cells also demonstrate their potential as a therapeutic strategy [96,97,98]. Even though there are numerous RA treatment options, we still lack a reliable tool to predict a particular medication’s effects. More interventional trials are needed in this subject to learn more about the effectiveness, adverse effects, and impact on quality of life. Table 1 represents the most emerging therapies and their respective clinical trials phases.

## 5. Conclusions

The literature suggests that MTX remains the first choice to treat RA among DMARDs and is sometimes used with a combination of hydroxychloroquine, leflunomide, and sulfasalazine; however, it is sometimes not sufficient. TNFi such as adalimumab, etanercept, infliximab, Janus kinase inhibitors (JAK inhibitors), and abatacept can be combined with conventional DMARDs or monotherapy if MTX fails to alleviate the disease progression. Nanoparticles, iguratimod, and fenebrutinib may have promising benefits for a patient for whom traditional therapy is ineffective. Toll-like receptor-4 targets, NLRP3 targets, and mesenchymal stem cell therapies can be cost-effective alternatives that further revolutionize RA treatment. Long-term treatment is recommended to archive the desired response to therapy, even with the rapid development of new RA drugs. Regular monitoring is also required when the patient is on DMARDs therapy. The bioinforma0tic analysis by STRING-MODEL showed no interactions between MTX targets including TYMS, ATIC, and DHFR genes and other drugs. This means an undiscovered molecular mechanism which should be associated with the MTX signaling pathway with the different pathways in which combined drugs are involved. This finding may shed light on future studies of RA and drug administration. Also, this suggestion may result in drug discovery, and the network of protein-coding genes involved in the treatment of RA can be a reference to design new NGS-based panels for personalized medicine via WGS and WES.

## Figures and Tables

**Figure 1 jpm-13-01550-f001:**
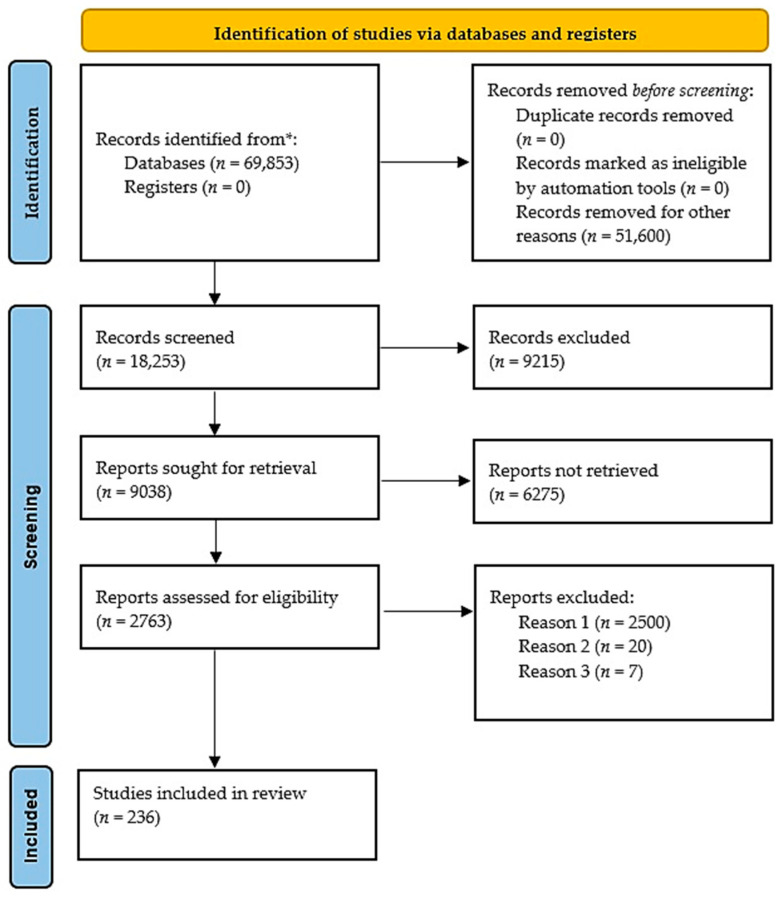
The PRISMA diagram of Rheumatoid Arthritis (RA) treatment in the current systematic study that represents the final included 236 related papers. Reason 1 was due to the lack of covering MTX in combination with other drugs; Reason 2 was the repeated reports; and Reason 3 was the papers with unclear results.

**Figure 2 jpm-13-01550-f002:**
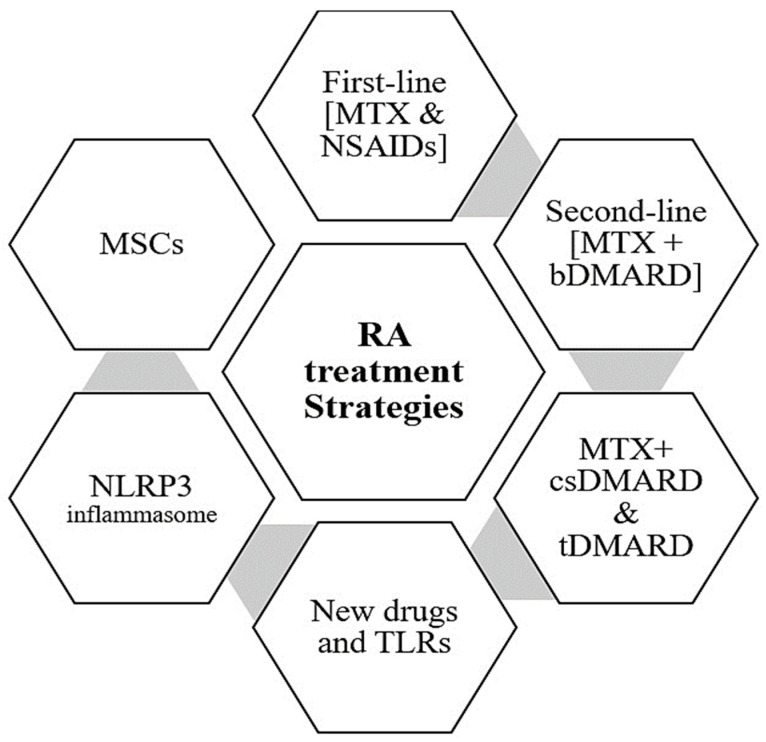
Rheumatoid Arthritis (RA) treatment strategies including first-line treatment Methotrexate (MTX) and Nonsteroidal anti-inflammatory drugs (NSAIDs) such as acetylsalicylate (Aspirin), naproxen (Naprosyn), ibuprofen (Advil and Motrin), and etodolac which are instances of fast-acting medications; the second-line treatment disease-modifying anti-rheumatic drugs (DMARDs) such as hydroxychloroquine, and sulfasalazine; biologic medications in combination with MTX like etanercept, infliximab, adalimumab, golimumab, and certolizumab pegol, anakinra, and rituximab; new drugs such as mavrilimumab, iguratimod, upadacitinib, and fenebrutinib; Toll-like receptors (TLRs); nucleotide-binding domain and leucine-rich repeat pyrin containing protein-3 (NLRP3) inflammasome; and finally, mesenchymal stem cells (MSCs).

**Figure 3 jpm-13-01550-f003:**
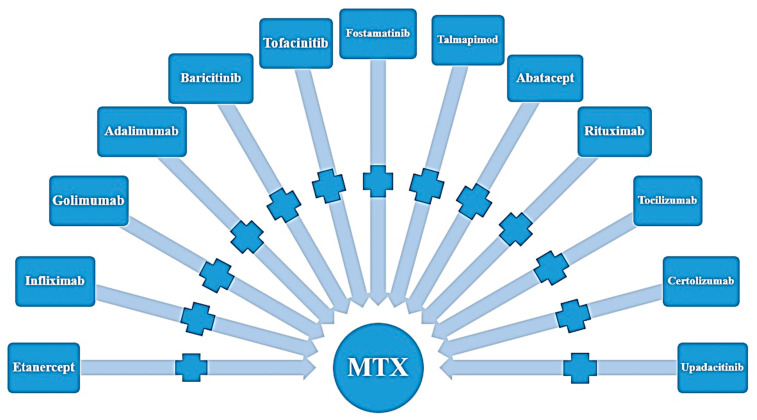
Shown are second-line treatments of Rheumatoid Arthritis (RA) by administrating MTX (MTX) in combination with other medications such as etanercept, infliximab, golimumab, adalimumab, bariceritinib, tofacitanib, fostamatinib, talmapimod, abatacept, tocilizumab, rituximab, certolizumab, and upadacitinib.

**Figure 4 jpm-13-01550-f004:**
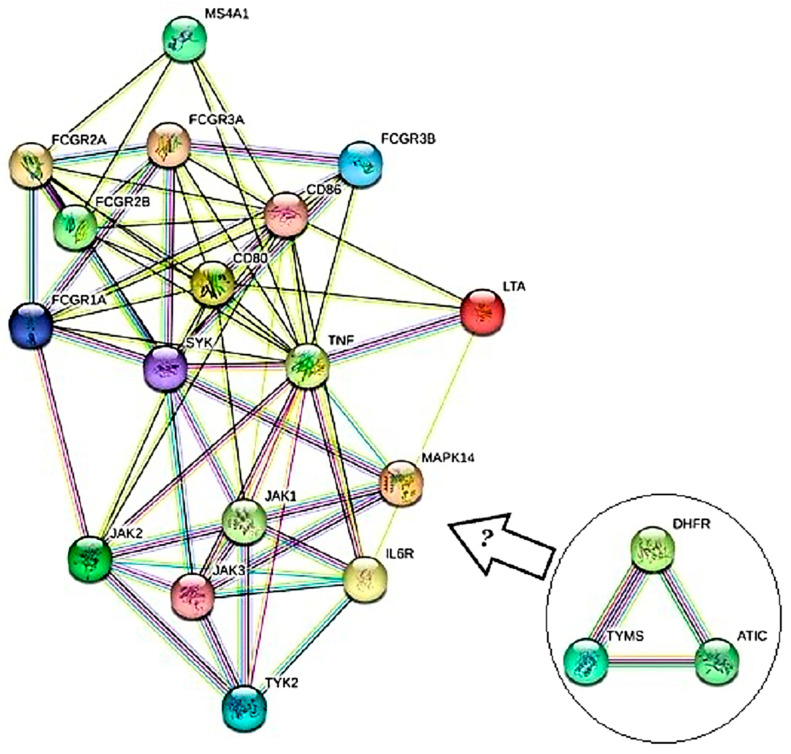
The new network was found by bioinformatics analyses based on DrugBank data and STRING-MODEL. The network illustrates the protein-coding targets of all MTX and other drugs in combination with MTX, including TYMS, ATIC, DHFR, TNF, LTA, FCGR1A, FCGR2A, FCGR2B, FCGR2C, FCGR3A, FCGR3B, JAK1, JAK2, JAK3, TYK2, SYK, MAPK14, CD80, CD86, IL6R, and MS4A1 genes. The question mark in the arrow refers to the unknown relationship of MTX with a combination of other drugs in this network.

**Table 1 jpm-13-01550-t001:** The most recent emerging therapeutic evidence for Toll-like 4 receptors, NLRP3 inflammasome, and mesenchymal stem cells.

Emerging Therapy	Author	Findings	Year	Ref.
TLR4	Li et al.	Rosmanol and Carnosol synergistically alleviate RA by inhibiting TLR4/NF-κB/MAPK pathway	2021	[105]
TLR4	Li et al.	A novel drug combination of mangiferin and cinnamic acid alleviates RA by inhibiting TLR4/NFκB/NLRP3 activation-induced pyroptosis	2022	[106]
NLRP3 inflammasome	Werner and Wagner	Increased extracellular Ca^2+^, calciprotein particles, and pro-inflammatory cytokines drive a vicious cycle of inflammation and bone destruction which in turn offers new potential therapeutic approaches.	2023	[107]
NLRP3 inflammasome	Li et al.	NLRP3 gene polymorphisms may play a role in the pathogenesis of RA and primary SS. The T allele of rs4612666 CT increased the susceptibility to RA disease.	2023	[108]
NLRP3 inflammasome	Liu et al.	The anti-inflammatory and antirheumatic effect of notopterygium may involve regulating NLRP3 inflammasome activation through mitochondria and NLRP3 is probably the key target molecule of notopterygium in the treatment of RA.	2023	[109]
NLRP3 inflammasome	Zhao et al.	Regulatory effect of zinc finger protein A20 on rheumatoid arthritis through NLRP3/Caspase-1 signaling axis mediating pyroptosis of Human Fibroblast-Like Synoviocytes (HFL)-RA cells	2023	[110]
NLRP3 inflammasome	Sun et al.	T-cell activation Rho GTPase activating protein (TAGAP) activates Th17 cell differentiation by promoting RhoA and NLRP3 to accelerate rheumatoid arthritis development	2023	[111]
NLRP3 inflammasome	Ye et al.	Sulforaphene targets NLRP3 inflammasome to suppress M1 polarization of macrophages and inflammatory response in rheumatoid arthritis	2023	[112]
NLRP3 inflammasome	Jiang et al.	Osthole (OST), a characteristic coumarin compound which is demonstrated as a potential AMPK agonist that inhibits NLRP3 inflammasome activation by regulating mitochondrial homeostasis for combating rheumatoid arthritis	2023	[113]
NLRP3 inflammasome	Zhang et al.	acetyltransferase KAT2A licenses metabolic and epigenetic reprogramming for NLRP3 inflammasome activation in inflammatory macrophages, thereby targeting KAT2A represents a potential therapeutic approach for patients suffering from RA.	2023	[114]
Mesenchymal stem cells	Elbasha et al.	Effect of autologous bone marrow-derived mesenchymal stem cells in treatment of rheumatoid arthritis	2023	[115]
Mesenchymal stem cells	Khorashad et al.	A significant change in the gene expression of TGFB1 and IFNG was consistent with the amelioration of clinical manifestations, suggesting a mechanism of action for MSCs in the treatment of RA.	2023	[116]
Mesenchymal stem cells	Choi et al.	Exosomes derived from mesenchymal stem cells primed with disease-condition-serum improved therapeutic efficacy in a mouse rheumatoid arthritis model via enhanced TGF-β1 production	2023	[117]
Mesenchymal stem cells	Zeng et al.	The effects of IL-1β stimulated human umbilical cord mesenchymal stem cells on polarization and apoptosis of macrophages in rheumatoid arthritis	2023	[118]
Mesenchymal stem cells	He et al.	Curcumin-loaded mesenchymal stem cell-derived exosomes efficiently attenuate proliferation and inflammatory response in rheumatoid arthritis fibroblast-like synoviocytes	2023	[119]
Mesenchymal stem cells	Ren et al.	Genetically engineered MSC-derived exosomes as a potential regulator of Th1 cell response in rheumatoid arthritis	2023	[120]
Mesenchymal stem cells	Rui et al.	A promising PD-L1 expression was identified on the exosomes, which potently suppressed Tfh cell polarization via inhibiting the PI3K/AKT pathway.	2023	[121]

The abbreviations are as follows: Ref: reference, TLR4: Toll-like Receptor 4, RA: Rheumatoid Arthritis, MSC: Mesenchymal Stem cell.

## Data Availability

All relevant clinical data are presented in this article. The raw data can be made available upon request to the first author.

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
