# Peer review of "Network-Based In Silico Analysis of New Combinations of Modern Drug Targets with Methotrexate for Response-Based Treatment of Rheumatoid Arthritis"

_jpm, 2023, doi:10.3390/jpm13111550_

Round 1

Reviewer 1 Report

Comments and Suggestions for Authors This is a really interesting review describing all the treatment arsenal, including the emerging therapies, for RA patients. It's well written but some conditions should be acknowledged.

I have written some suggestions in the manuscript. Please carefully review the manuscript. Introduction section:

- Describe the pathogenesis of RA (innate and adaptive immune responses, proinflammatory cytokines involved...) and add an illustrative image. This will help to understand the drug mechanisms of action.

- Correct the reference numbers.

- Leflunomide is not a biological agent. It's a conventional synthetic DMARD (csDMARD), so it must be described right after methotrexate. 
- Include a description of the side effects of methotrexate.

- Avoid using the brand names of the drugs, instead use generic names.

- Add the others JAK inhibitors (such as baricitinib and upadacitinib, both approved and used in the clinical practice)

Methods

- Upadacitinib shouldn't be included in the new drugs. It's approved and largely used in RA patients. 

Results

- I don't think table 1 adds relevant information. I suggest deleting it.

- Before describing the others bDMARD and tDMARD, I think you should mention that TNFi was the first class of bDMARD approved  for the treatment of RA and discuss the positive impact on patients' quality of life.

- I don't agree with the scheme of figure 2.The second line of therapy already includes MTX + biologics (bDMARD) and not DMARD (it's a vague term that includes csDMARD, bDMARD, tDMARD..). I suggest changing to:

1. csDMARD 
2. bDMARD
3. tDMARD
4. New drugs and TLR
5. NLRP3
6. MSCs

- Add a table detailing emerging therapies and their respective clinical trials phases.

- In the text you describe. "Figure 2 depicts the most prevalent adverse side effects associated with current DMARDS " but the figure 2 don't describe the adverse side effects. Add the correct figure.
- In figure 3, add certolizumab pegol and upadacitinib. 

Discussion

- Incorporate a reference to the bioinformatics analysis in the discussion or conclusion. I think you should integrate the first part of the conclusion in the discussion. Start by writing that MTX is the first line of treatment, but sometimes its not sufficient. Then, discuss a little about the others approved bDMARD (like you have done in conclusion). After that, talk about the patients difficult to treat and the need for new drugs. Finally, talk about the results of these new drugs.

Author Response

Reviewer 1 Reply

This is a really interesting review describing all the treatment arsenal, including the emerging therapies, for RA patients. It's well written but some conditions should be acknowledged.

I have written some suggestions in the manuscript. Please carefully review the manuscript. Introduction section:

-Describe the pathogenesis of RA (innate and adaptive immune responses, proinflammatory cytokines involved...) and add an illustrative image. This will help to understand the drug mechanisms of action.

Reply: The pathogenesis of RA including innate and adaptive immune responses are now added to the introduction section with highlights. Due to the strategy of the paper and not being the main focus of the paper, we think that the honorable reviewer will relinquish it.

- Correct the reference numbers.

Reply: The references are corrected now and highlighted.

- Leflunomide is not a biological agent. It's a conventional synthetic DMARD (csDMARD), so it must be described right after methotrexate. 

Reply: It is corrected now.

-Include a description of the side effects of methotrexate.

Reply: It is included now with highlight.

- Avoid using the brand names of the drugs, instead use generic names.

Reply: The generic brand names of the drugs are now deleted.

- Add the others JAK inhibitors (such as baricitinib and upadacitinib, both approved and used in the clinical practice)
Reply: They are now added with highlights.

Methods

- Upadacitinib shouldn't be included in the new drugs. It's approved and largely used in RA patients. 

Reply: It is deleted now.

Results

- I don't think table 1 adds relevant information. I suggest deleting it.

Reply: Table 1 is deleted now.

- Before describing the others bDMARD and tDMARD, I think you should mention that TNFi was the first class of bDMARD approved  for the treatment of RA and discuss the positive impact on patients' quality of life.

Reply: It is mentioned now and highlighted.

- I don't agree with the scheme of figure 2.The second line of therapy already includes MTX + biologics (bDMARD) and not DMARD (it's a vague term that includes csDMARD, bDMARD, tDMARD..). I suggest changing to:

1. csDMARD 
2. bDMARD
3. tDMARD
4. New drugs and TLR
5. NLRP3
6. MSCs

Reply: It is corrected now,

- Add a table detailing emerging therapies and their respective clinical trials phases.

Reply: Table 1 is added based on the reviewer’s comment.

- In the text you describe. "Figure 2 depicts the most prevalent adverse side effects associated with current DMARDS " but the figure 2 don't describe the adverse side effects. Add the correct figure.

Reply: This is deleted now.

- In figure 3, add certolizumab pegol and upadacitinib. 

Response: They are added now.

Discussion

- Incorporate a reference to the bioinformatics analysis in the discussion or conclusion. I think you should integrate the first part of the conclusion in the discussion. Start by writing that MTX is the first line of treatment, but sometimes its not sufficient. Then, discuss a little about the others approved bDMARD (like you have done in conclusion). After that, talk about the patients difficult to treat and the need for new drugs. Finally, talk about the results of these new drugs.

Response: These suggestions are now added and changes are highlighted.

Reviewer 2 Report

Comments and Suggestions for Authors

The manuscript describes a network-based in silico analysis of RA therapy.

This is an interest paper particularly for its employment of the unusual analysis methods. However, there are some revisions that should be done to improve the presentation of the paper.

1. Please revise the result section of the abstract to include more specific results, rather than a general statement.

2. Conclusion is too lengthy, so it should be revised to be more specific to the paper's key findings and what impact it may pose to the current paradigm of RA therapy.

3. Methods: please specify the key words used for the database search and list the people who review abstracts and how disagreements are resolved. This is to ensure that the search strategy is clear and repeatable.

Comments on the Quality of English Language

no specific comment

Author Response

The manuscript describes a network-based in silico analysis of RA therapy.

This is an interest paper particularly for its employment of the unusual analysis methods. However, there are some revisions that should be done to improve the presentation of the paper.

  1. Please revise the result section of the abstract to include more specific results, rather than a general statement.

Response: We tried our best to add more specific results which you can find in highlight, but the limitation of Abstract does not let further words.

  1. Conclusion is too lengthy, so it should be revised to be more specific to the paper's key findings and what impact it may pose to the current paradigm of RA therapy.

Response: The conclusion is now shortened and revised and the suggestions in the conclusion will pose to the current paradigm of RA therapy.

  1. Methods: please specify the key words used for the database search and list the people who review abstracts and how disagreements are resolved. This is to ensure that the search strategy is clear and repeatable.

Response: Please see the first paragraph of Material and methods section; which we clearly stated the keywords and the strategy of the review. Pleas see “Following PRISMA guidelines [1], this review systematically searched Rheumatoid Arthritis in PubMed (https://pubmed.ncbi.nlm.nih.gov/ ) and found 69,853 published papers and books from 1876 to 2023. By searching Rheumatoid Arthritis drugs, the results decreased to 18,253 entries from 1947 to 2023…. Only one paper suggested NSAIDs as the first-line RA therapy”.

As we mentioned in the authors contribution part, AS: Alireza Sharafshah performed the methodology and it is completely repeatable if anyone wants to test it.

Round 2

Reviewer 2 Report

Comments and Suggestions for Authors

Thank you for the revision. I have no further comments.